# H3K4me3 Is a Potential Mediator for Antiproliferative Effects of Calcitriol (1α,25(OH)2D3) in Ovarian Cancer Biology

**DOI:** 10.3390/ijms21062151

**Published:** 2020-03-20

**Authors:** Nan Han, Udo Jeschke, Christina Kuhn, Anna Hester, Bastian Czogalla, Sven Mahner, Miriam Rottmann, Doris Mayr, Elisa Schmoeckel, Fabian Trillsch

**Affiliations:** 1Department of Obstetrics and Gynaecology, University Hospital, LMU Munich, Marchioninistr. 15, 81377 Munich, Germany; N.Han@med.uni-muenchen.de (N.H.); christina.kuhn@med.uni-muenchen.de (C.K.); anna.hester@med.uni-muenchen.de (A.H.); bastian.czogalla@med.uni-muenchen.de (B.C.); sven.mahner@med.uni-muenchen.de (S.M.); 2Department of Obstetrics and Gynaecology, University Hospital Augsburg, Stenglinstr. 2, 86156 Augsburg, Germany; 3Munich Cancer Registry (MCR), Bavarian Cancer Registry—Regional Center Munich (LGL), Institute for Medical Information Processing, Biometry and Epidemiology (IBE), Ludwig-Maximilians-University (LMU), 81377 Munich, Germany; rottmann@ibe.med.uni-muenchen.de; 4Department of Pathology, LMU Munich, Thalkirchner Str. 36, 80337 Munich, Germany; doris.mayr@med.uni-muenchen.de (D.M.); elisa.schmoeckel@med.uni-muenchen.de (E.S.)

**Keywords:** ovarian cancer, histone 3 lysine 4 trimethylation (H3K4me3), histone modification, calcitriol, 1α,25(OH)_2_D_3_, prognosis, vitamin D receptor, cell proliferation

## Abstract

Posttranslational histone modification plays an important role in tumorigenesis. Histone modification is a dynamic response of chromatin to various signals, such as the exposure to calcitriol (1α,25(OH)_2_D_3_). Recent studies suggested that histone modification levels could be used to predict patient outcomes in various cancers. Our study evaluated the expression level of histone 3 lysine 4 trimethylation (H3K4me3) in a cohort of 156 epithelial ovarian cancer (EOC) cases by immunohistochemical staining and analyzed its correlation to patient prognosis. The influence of 1α,25(OH)_2_D_3_ on the proliferation of ovarian cancer cells was measured by BrdU proliferation assay in vitro. We could show that higher levels of H3K4me3 were correlated with improved overall survival (median overall survival (OS) not reached vs. 37.0 months, *p* = 0.047) and identified H3K4me3 as a potential prognostic factor for the present cohort. Ovarian cancer cell 1α,25(OH)_2_D_3_ treatment induced H3K4me3 protein expression and exhibited antiproliferative effects. By this, the study suggests a possible impact of H3K4me3 expression on EOC progression as well as its relation to calcitriol (1α,25(OH)_2_D_3_) treatment. These results may serve as an explanation on how 1α,25(OH)_2_D_3_ mediates its known antiproliferative effects. In addition, they further underline the potential benefit of 1α,25(OH)_2_D_3_ supplementation in context of ovarian cancer care.

## 1. Introduction

Epithelial ovarian cancer (EOC) is one of the most common malignancies in women, with the highest mortality and a five-year survival rate of less than 45% [1,2]. The main reasons for poor prognosis are the lack of effective screening methods and the late clinical manifestation due to asymptomatic tumor progression in most cases. Primary surgical debulking and subsequent platinum-based chemotherapy is currently the mainstay of treatment with still a curative intention for advanced ovarian cancer. Recently, antiangiogenic treatment and poly (ADP-ribose) polymerase (PARP) inhibitors could be added as targeted therapies to first-line treatment with significant improvement of progression-free survival (PFS) [3]. However, molecular markers are still missing to tailor systemic treatment and reliable predictors from biologic specimens have not yet been fully elucidated.

Global changes in the epigenetic landscape are one important hallmark of cancer. Posttranslational histone modification is considered as a common phenomenon in tumor progression and one of the earliest events in carcinogenesis [4]. Recently, histone modification patterns have been identified as useful in distinguishing subtypes of cancer patients with distinct clinical outcomes, thereby expanding prognostic capabilities [5]. Heterogeneity in cellular (such as global or bulk) levels of histone modifications can be detected by immunohistochemistry (IHC) assay at the level of whole nuclei of cancer cells in tissue specimens [6]. Meanwhile, immunocytochemistry (ICC) is also applied to confirm histone H3 modification expression in normal and cancer cells [5,7]. Former studies have reported that alteration in the histone modification patterns can provide prognostic information for several cancers, including those detected in colon [8,9], kidney [10,11], lung [5,12], stomach [13], pancreas [14,15], ovary [14], and breast [14,16].

Histone H3 tri methyl K4 (H3K4me3) is one of the most extensively studied patterns of histone modifications, which either contributes to transcription activation or is associated with suppressed gene expression [17,18]. Previous studies have proven prognostic value of H3K4me2/3 for colon cancer [8], renal cell carcinoma [10], and lung and kidney cancer [5]. In ovarian cancer biology, prior studies evaluated the expression and role of H3K4me3 protein indirectly via examining gene sets associated with H3K4me3 marks at transcription start sites [19] or via detection of its methyltransferase and demethyltransferase [20,21]. However, the prognostic significance of H3K4 trimethylation and ovarian cancer remains unclear for now.

Gene regulation mediated by nuclear receptors via chromatin remodelling and histone-modifying complexes is one example of how posttranslational changes may influence tumor growth [22]. A well-characterized example of histone modification mediated by nuclear receptors is that 1α,25(OH)_2_D_3_, known as calcitriol or active form of vitamin D, can regulate histone modification and can inhibit cancer progression through the vitamin D receptor (VDR) [23,24]. Recently, it was reported that 1α,25(OH)_2_D_3_ sensitizes the tumor suppressor p16 in kidney cancer cell lines [25]. Another study suggested that 1α,25(OH)_2_D_3_ induces the expression of histone demethylase JMJD3, thus enhancing trimethylated H3K4 elevated by 1α,25(OH)_2_D_3_ at several target gene promoters in breast cancer epithelial cells [26].

As recently reported, different histone modifications are linked with chemotherapy resistance and become an emerging fields of chemotherapeutic targets [27]. However, for EOC, there is only limited evidence of a relation between histone modification expression and development of platinum resistance to date. Prior analyses have demonstrated that the acquired platinum-resistant cell line PEO4 had significantly different H3K4me3 expression compared to the chemosensitive cell line PEO1 [19]. In another report, there was no difference in H3K4me3 expression. However, H3K4me3 proteins could be suppressed by Trichostatin A and 5-aza-CdR in a A2780/A2780cis cisplatin-resistance cell line model [28].

In order to better understand the prognostic value of H3K4me3 for EOC, we correlated H3K4me3 expression in EOC specimens with their clinical course. Alterations in H3K4me3 expression and cell proliferation following 1α,25(OH)_2_D_3_ treatment in ovarian cancer cell lines were also explored to put the 1α,25(OH)_2_D_3_ expression in the biologic context.

## 2. Results

### 2.1. H3K4me3 Staining in EOC Patients

Primary EOC specimens from a total of 156 patients with a median age of 64.0 years (range 33–100 years) and a median follow up of 33.5 months were studied. The median of progression-free survival (PFS) was 22.8 months with a corresponding overall survival (OS) of 40.9 months (range: 0–230.0 months). Of this cohort, a total of 91.7% (142 of 156) samples showed nuclear staining of H3K4me3, while the negative cases and missing cases due to technically failure accounted for 3.2% (5 of 156) and 5.1% (8 of 156), respectively. Among all the positive H3K4me3 staining samples, median Immunoreactive Score (IRS) was 6 (23%, 34 of 148). More specifically, 24 (16.2%) samples were identified with strong immunoreactivity (IRS = 9–12), while 24 (16.2%) samples and 95 (64.2%) samples exhibited weak staining (IRS = 2–3) and moderate staining (IRS = 4–8) (Figure 1). No significant correlation of H3K4me3-expression clinical and pathological parameters was detected (Table 1).

### 2.2. High H3K4me3 Expression Was Associated with Increased Overall Survival in EOC Patients

We analyzed the correlation between H3K4 trimethylation levels and patient outcomes. As shown in the Kaplan–Meier curve, patients with high expression of H3K4me3 (IRS = 9–12) had improved median overall survival compared to patients with lower levels (median OS not reached vs. 37.0 months, *p* = 0.047, hazard ratio = 0.52, 95% confidence interval = 0.47–0.57) (Figure 2).

### 2.3. Cox Regression

The multivariate Cox regression analysis of accepted prognostic factors indicated that grading and FIGO stage were independent prognostic factors for the present cohort while H3K4me3 exhibited borderline significance (Table 2).

### 2.4. Co-Expression of VDR and H3K4me3 Protein in Ovarian Cancer Patient Tissue

We further examined the co-expression of VDR and H3K4me3 in ovarian cancer tissues. Double-immunofluorescence in ovarian cancer patients’ tissues revealed that H3K4me3 protein is co-localized with VDR. While Histone H3 tri methyl K4 was present in the nuclei, VDR was mainly detected in the cytoplasm (Figure 3).

### 2.5. 1α,25(OH)_2_D_3_ Induced H3K4me3 Expression in A2780 and A2780cis Cell Lines

According to the results of immunocytochemistry (ICC) in epithelial ovarian cancer cell lines, A2780 cells displayed strongly positive immunostaining of H3K4me3 following treatment of 1000 nM 1α,25(OH)_2_D_3_ for 24 h and 48 h (Figure 4A3,B3). The mean optical density (OD) values of nuclear H3K4me3 labeling increased more significantly than in control and lower concentration 1α,25(OH)_2_D_3_ groups (*p* < 0.05 or *p* < 0.01, Figure 4C). Accordingly, staining in A2780 cells (treated by 100 nM 1α,25(OH)_2_D_3_ for 48 h) was higher than in the controls (*p* < 0.05, Figure 4B2,C), but there was no significant change of H3K4me3 expression in the cells treated with 100 nM 1α,25(OH)_2_D_3_ for 24 h (Figure 4A2,C, *p* > 0.05).

In A2780cis, strongly positive immunostaining was observed in cells treated with 1000 nM 1α,25(OH)_2_D_3_ for 24 h and 48 h (Figure 5A3,B3) and the mean OD value was significantly higher than in the control group (Figure 5C, *p* < 0.01). A2780cis cells receiving 100 nM 1α,25(OH)_2_D_3_ treatment for 48 h displayed no change compared with control (Figure 5C, *p* > 0.05); however, weakly positive staining was visible (Figure 5A2,B2).

### 2.6. Decreased Proliferation of Ovarian Carcinoma Cells by 1α,25(OH)_2_D_3_

Results of the BrdU assays carried out in 1α,25(OH)_2_D_3_-treated cells and control cells indicate that the growth of A2780 cells treated with 100 nM 1α,25(OH)_2_D_3_ is inhibited after 48 h (*p* < 0.05), while no significant difference was observed between the untreated control cells and treated cells in the 24 h group (*p* = 0.384). The inhibitory effects on cell proliferation were also observed in the A2780 cell lines exposed to 1000 nM 1α,25(OH)_2_D_3_ (Figure 6, *p* ≤ 0.005).

Among the platinum-resistant A2780cis cells treated with 1000 nM 1α,25(OH)_2_D_3_, a growth-promoting effect can be seen in the 24 h group (*p* < 0.05), while the proliferation was inhibited after 48 h (*p* < 0.01). No effects were observed in the A2780cis cells treated with lower concentration (100 nM) of 1α,25(OH)_2_D_3_ (Figure 6, both *p* > 0.05).

## 3. Discussion

Within the current analysis, we could demonstrate that a high-level expression of H3K4me3 is associated with better prognosis in EOC patients. In functional studies with ovarian cancer cell lines A2780 and A2780cis, ICC testing revealed that treatment with 1α,25(OH)_2_D_3_ can induce a dose-dependent H3K4me3 expression in ovarian cancer cell lines. Following high-dose 1α,25(OH)_2_D_3_-treatment, cell proliferation was inhibited in A2780 and A2780cis cell lines underlining the functional significance of this pathway.

Accumulating evidence suggests that histone posttranslational modifications (PTMs) play a crucial role in many key cellular processes including gene transcription, DNA replication, and reparation through alterations in chromatin structure [29] and that aberrant histone modifications could cause various diseases [30,31]. Unlike genetic alterations, changes in histone modifications are reversible and match the dynamic chromatin in nature. According to the “histone code hypothesis”, histone modifications can make considerable impact on chromosome function through distinct mechanisms [32]. For example, histone methyltranferase (classified as “writer”) catalyzes the transfer of methyl groups to lysine and arginine residues of histone proteins and histone demethylase (classified as “eraser”) removes methyl groups from histone protein [33,34,35]. Some of the modifications like H3K4me2/3, H3K36me, and H3K79 were associated with “open” chromatin and active chromatin, whereas others such as H3K9me, H3K27me, and H4K20me are related to “closed” chromatin and transcriptional repression [36,37]. As a prominent example of the histone modifiers, we have chosen H3K4me3 for the detailed clinical and biologic evaluation in EOC.

The present H3K4me3 expression analysis elucidated that high expression levels are correlated with better clinical outcomes being in agreement with previous observation of the prognostic significance of H3K4me3 in renal cell carcinoma [10]. Kumar et al. reported that H3K4me3 level was significantly decreased in metastasis of renal cell carcinoma and therefore suggested the implication as a biomarker to discriminate metastatic from nonmetastatic tumors [11]. In contrast, other reports suggested that increased H3K4me3 expression was associated with impaired overall survival in various cancers, such as hepatocellular carcinoma [38], cervical cancer [39], and early-stage colon cancer [8]. The discrepancy could be due to the distinct distribution of H3K4me3 expression in various kinds of cancers and even in different stages and histological subtypes of a specific tumor entity. As member of the transcription factor family, VDR can dynamically interact with chromatin components and can therefore potentially mediate the effects of the histone modifications [40]. Therefore, we performed co-immunofluorescence of H3K4me3 and VDR in selected ovarian cancer specimens. Our result displayed obvious co-expression of VDR and H3K4me3 in ovarian cancer. A former study has shown that the level of histone modifications (including H3K4me3) is significantly modulated via enhancing genome-wide the rate of accessible chromatin and vitamin D receptor (VDR) binding by 1,25(OH)_2_D_3_ stimulation [24]. A recent study suggested that the interplay between H3K4 methyltransferase MLL1 and vitamin D pathway could determine cell fate in vitro [41]. Our observation in ovarian cancer patients’ tissues coincide with previous studies. Antiproliferative effects of 1α,25(OH)_2_D_3_ may involve the mechanisms associated with apoptotic pathway activation and angiogenesis inhibition [42], and the vitamin D receptor was proposed to be crucial for tumor suppression [43]. To further understand the impact of 1α,25(OH)_2_D_3_ on the expression of H3K4me3 and cell proliferation, immunocytochemistry and BrdU assay were carried out in ovarian cancer cell lines. Trimethylated H3K4 is a biomarker for transcription initiation and elongation [18]. In the absence of ligands, VDR was shown to interact with corepressor proteins and chromatin-modifying enzymes like histone deacetylase (HDACs) in the deactivation phase [40]. In the activation phase, binding of 1α,25(OH)_2_D_3_ leads to alterations in the receptor conformation and access to the binding of co-activators that have histone acetylase activity or are complexed with proteins harboring such activity [40]. Here, we found that 1α,25(OH)_2_D_3_ could induce an increased expression of H3K4me3 protein in both the A2780 and A2780cis cell lines and therefore irrespective of the response to platinum treatment.

The Wnt pathway plays an important role in the carcinogenesis of all ovarian cancer subtypes including ovarian cancer stem cells (CSCs) [44] and is considered to promote cancer progression as well as chemoresistance between parental A2780 and platinum-resistant A2780cis cell lines [45,46]. Additionally, some findings indicated that epithelial ovarian cancers may derive from a subpopulation of CD44^+^CD117^+^ and that drug-resistant A2780 cells display ovarian CSC properties [47,48]. It has been reported that calcitriol (1α,25(OH)_2_D_3_) can deplete the ovarian CSCs characterized by CD44^+^CD117^+^ by targeting the Wnt signaling pathway [49]. In a former study, DACT3 (a negative regulator of Wnt/β-catenin pathway) could inhibit Wnt/ß-catenin activity although the activating mark H3K4me3 remained at high levels near the DACT3 transcription start site in colorectal cancer cells [50]. Taken together, these findings could partly explain why A2780 as well as A2780cis had a comparable reaction to 1α,25(OH)_2_D_3_ treatment with an induced H3K4me3 expression. However, the correlation and the relevance of H3K4me3 to Wnt pathway in ovarian cancer cells will require further research in the future.

Additionally, our results were consistent with findings from Goeman et al., who suggested that 1α,25(OH)_2_D_3_ increased the trimethylation of H3K4 at target gene promoters, and the expression of the targeted gene in breast cancer cells was upregulated after treatment with 1α,25(OH)_2_D_3_ [26]. Menin, which is a putative tumor suppressor and an integral part of MLL1 and MLL2 histone methyltransferase complexes [51,52], has been suggested to directly interact with VDR and to enhance the transcriptional activity of the receptor [53]. Therefore, histone methyltransferases of H3K4me3 might be absorbed by activated VDR, thus increasing the level of trimethylated H3K4.

Our study indicates that 1α,25(OH)_2_D_3_ has antiproliferative activity on ovarian cancer cells, irrespective from the resistance pattern, mainly at a higher dose for a longer treatment period. This appears in line with the observations found in OVCAR-3 cell line [54], indicating the impact of 1α,25(OH)_2_D_3_ on cell proliferation varied by concentration and treatment time. A high dosage of 1α,25(OH)_2_D_3_ decreased the proliferative activity through G_0_/G_1_ arrest [55] and upregulation of the cyclin-dependent kinase inhibitor 1A (CDKN1A, known as p21) [56].

In A2780 and A2780cis cells, strongly increased expression of H3K4me3 was accompanied by inhibited activity of 1α,25(OH)_2_D_3_, which is in line with the observation that only upregulated genes after 1α,25(OH)_2_D_3_ treatment show a concurrent significant increase of H3K4me3 at their transcriptional start site [26].

Based on the results of the current study, we assume that H3K4me3 has a pivotal role in mediating the already accepted antiproliferative ability of 1α,25(OH)_2_D_3_ in ovarian cancer biology. The demonstrated relation between H3K4me3 and 1α,25(OH)_2_D_3_ could explain how calcitriol exhibits its effects on tumor suppression and underlines the potential benefit of calcitriol supplementation in context of ovarian cancer care.

## 4. Materials and Methods

### 4.1. Patients and Tissue Microarray

The tissue microarray was conducted with 156 EOC tissue specimens obtained from patients who underwent surgery for EOC in the Department of Obstetrics and Gynecology of the Ludwig-Maximilians-University Munich between 1990 and 2002. Clinical data was derived from patient charts and follow-up data was obtained from Munich Cancer Registry. All samples were prepared by formalin fixation and paraffin embedding (FFPE). Three representative tissues were taken from each patient for the microarray analysis to obtain a more accurate image of EOC.

### 4.2. Ethics Approval

All epithelial ovarian cancer specimens were derived from the archives of the Department Gynecology and Obstetrics in LMU Munich, which were initially applied for pathological diagnostics. In all cases, the diagnostic procedures were completed before the current study was performed. Our study was approved by the Ethics Committee of the Ludwig-Maximilians-University (Date: 30 September 2009; approval number: 227-09; Munich, Germany). All experiments in this study were conducted in accordance with the Declaration of Helsinki. The authors were blind to patient information throughout the trial.

### 4.3. Immunohistochemistry

The paraffin-embedded and formalin-fixed samples from 156 EOC patients were used to construct a tissue microarray (TMA). Sections of 3 μm were cut from the TMA block and prepared for immunohistochemical (IHC) staining. Deparaffinization was conducted by using xylene, and the samples were rehydrated with ethanol at a descending concentration gradient. Endogenous peroxidase was quenched with 3% hydrogen peroxide in methanol at room temperature. The sections were placed in citrate buffer (pH = 6.0) and heated for 5 min at boiling temperature in a pressure cooker to retrieve the antigen. After cooling for 5 min, the sections were washed with distilled water and phosphate buffered saline solution (PBS) buffer. Appropriate blocking solution was applied to avoid nonspecific binding of immunoglobulins on one side to cell membranes or fatty tissue on the other side due to electrostatic charge. Afterwards, primary antibody H3K4me was applied and incubated overnight at 4 °C.

Growth immunohistochemical staining was performed by using post-block reagent and horse raddish peroxidase (HRP)-polymer, followed by substrate-staining with 3,3′-Diaminobenzidine (DAB). Subsequently, the sections were counterstained with haemalun for 2 min. Table 3 further presented details with regard to the suitable detective systems and corresponding steps. Ultimately, dehydration of the specimens was performed by using ethanol at an ascending concentration gradient. Tissues retrieved from colon and placenta were used as positive and negative controls in immunohistochemical staining.

Immunoreactive score (IRS) was used to evaluate the immunostaining results, which was calculated by multiplying the intensity of staining reaction (0 = no color reaction; 1 = weak reaction; 2 = moderate reaction; and 3 = intense reaction) by the percentage of positive cells (0 = 0%; 1 = 1–10%; 2 = 11–50%; 3 = 51–80%; and 4 = >80%). The calculated IRS ranged from 0 to 12, among which 0 indicated no expression of histone and 12 suggested strong expression of histone (Table 3). Positive as well as negative controls were included (Figure A1).

### 4.4. Double Immunofluorescence Staining

For the characterization of H3K4me3 and VDR expression in ovarian cancer, double immunofluorescence was applied by same the paraffin-embedded slides (*n* = 4). Paraffin-embedded slides (3 μm thick) were dewaxed in Roticlear for 20 min and washed in a descending ethanol series (100%, 70%, and 50%). Slides were heated in a pressure cooker using sodium citrated buffer (pH = 6.0), including 0.1 M citric acid and 0.1 M sodium citrate in distilled water. After cooling and washing in PBS buffer, slides were blocked with Ultra V Block (Lab Vision, Fremont, CA, USA) for 15 min at room temperature and then incubated with primary antibodies overnight at 4 °C. Both primary antibodies were diluted with a diluting medium (Dako, Hamburg, Germany) according to the following ratios: 1:100 for rabbit anti-Histone H3 tri methyl K4 IgG (Abcam, ab8580) and 1:100 for mouse anti-vitamin D receptor monoclonal IgG2a (Bio-Rad,MCA3543Z). After washing, slides were incubated with Alexa Fluor 488-/Cy3-labeled antibodies (Dianova, Hamburg, Germany) as fluorescent secondary antibodies for 30 min at room temperature in darkness. Alexa Fluor 488-and Cy3-labeled secondary antibodies were at dilutions of 1:100 and 1:500 in Dako, respectively. Finally, the slides were embedded in mounting medium for fluorescence with 4′,6-diamino-2-phenylindole (DAPI, Vectastain, Vector Laboratories, Burlingame, CA, USA) for blue staining of the nucleus after washing and drying. Digital photos were taken with a digital camera system (Axiocam; Zeiss CF20DXC; KAPPA Messtechnik, Gleichen, Germany) and digitally saved.

### 4.5. Cell Lines and Treatment

The human endometrioid ovarian carcinoma cell line A2780 and its platinum-resistant variant A2780cis were obtained from European Collection of Cell Cultures. The A2780 cell line was cultured in RPMI1640 (ThermoFisher Scientific, Waltham, MA, USA) with 10% fetal bovine serum. The A2780cis cells were continuously cultivated in presence of cisplatin at a concentration of 10 mg/mL. All cells were incubated in a humidified incubator at 37 °C and 5% CO_2_. Calcitriol was purchased from Cayman Chemical Company (Michigan, USA). Cells were treated with 0.1% Dimethyl sulfoxide (DMSO) as vehicle or calcitriol at indicated concentration in RPMI1640 with fetal bovine serum (FBS) (or with cisplatin).

### 4.6. Immunocytochemistry

Cells were inoculated into a four-well chamber slide (10^5^ cells/well) for immunocytochemistry (ICC) analysis. One day after cell inoculation, the medium was replaced by fresh RPMI1640/10% FBS containing vehicle or 1α,25(OH)_2_D_3_ at different concentrations (100 nM and 1000 nM) and the cells were incubated for indicated time periods (24 h and 48 h). After treatment, the slides were washed with PBS for 5 min and fixed with ice-cold 50%-methanol-50%-ethanol solution for 15 min at room temperature (RT). After the samples were air-dried, blocking solution was added to the slides and the cells were incubated for 5 min at RT, after which the blocking solution was drained away. The samples were incubated with anti-H3K4me3 (Abcam, ab8580) 100 μL/slides (1:500 dilution with PBS) for 16 h at 4 °C. Subsequently, the slides were placed in post-block solution for 20 min and HRP-polymer for 30 min at RT. After each session of incubation, the slides were washed in PBS for 5 min. Substrate-staining was performed with aminoethyl carbazole (AEC) for 4 min at RT, and the reaction was stopped in distilled water (Aqua Dest). Then, the counterstaining was carried out with haemalun for 30 s. Finally, the slides were placed in tap warm water and let sit for 4 min. Five visible fields of each immunocytochemistry staining slide were taken photos under a microscope (×40), and their optical density (OD value) was measured using Image J software v1.52p (National Institutes of Health, USA).

### 4.7. Cell Proliferation Assay

Quantification of cell proliferation was determined by the BrdU assay (Roche Applied Science, Mannheim, Germany) based on the measurement of a pyrimidine analogue (BrdU) incorporation during DNA synthesis. The experimental steps were carried out in accordance with the manufacturer’s instruction. Briefly, A2780 and A2780cis cell lines were inoculated in triplicate into 96-well flat-bottom plates at a density of 5000 cells/well and were treated with vehicle or 1α,25(OH)_2_D_3_ at indicated concentrations (100 nM and 1000 nM) [57] for different time periods (24 h and 48 h). After treatment, cells were labelled with BrdU and incubated for 2 h at 37 °C. After cell fixation, anti-BrdU-POD (100 μL/well) was applied and incubated for 1.5 h, followed by three times of washing with washing solution. Ten minutes after the substrate solution was added to each well, the reaction was stopped by adding 1 M H_2_SO_4_ (25 μL/well). The absorbance of the samples at the wavelength of 450 nm was determined by ELISA. All experiments were performed in triplicate.

### 4.8. Statistical Analysis

The nonparametric Mann–Whitney U test was adopted to assess the correlation between histone H3 tri methyl K4 scores and clinical outcomes. The Cox proportional hazard model was used for the multivariate analyses. The overall survival rate was analyzed by the Kaplan–Meier curve, and the difference in survival rate was tested by log-rank test. Mann–Whitney U test was also employed to calculate the statistical significance of OD values among different groups. Comparation of the absorbance values between the treated cell and controls was evaluated by paired-samples T test. A *p* value less than 0.05 was considered as statistically significant. All the statistical analyses were conducted with IBM SPSS 23 (Armonk, NY, USA), and plotting was completed with Graph-Pad Prism 8.02 (v8, La Jolla, San Diego, CA, USA).

## 5. Conclusions

In this study, we could demonstrate that high-level H3K4me3 expression is associated with improved outcome in patients with EOC. The results suggest that application of 1α,25(OH)_2_D_3_ increases the expression of H3K4me3 and exerts an inhibitory effect on cell proliferation in ovarian cancer cell lines. Therefore, the results may serve as an explanation on how calcitriol exhibits its effects on tumor suppression and underlines the potential benefit of calcitriol supplementation in context of ovarian cancer care.

## Figures and Tables

**Figure 1 ijms-21-02151-f001:**
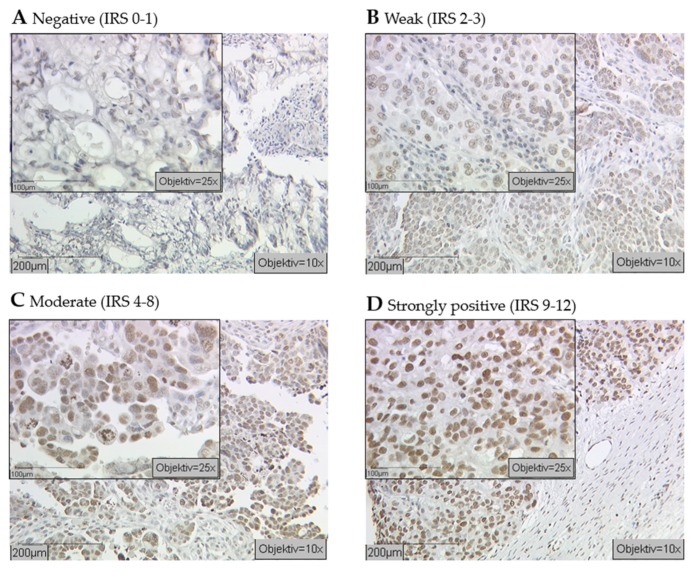
Cellular epigenetic heterogeneity in cancer: Immunohistochemical examination of ovarian cancer tissues with an antibody against histone Histone H3 tri methyl K4 (H3K4me3) revealed different expression levels (indicated by brown staining). Specimens were attributed to negative (**A**), weak (**B**), moderate (**C**), and strongly positive (**D**) expression levels of H3K4me3 (scale bar 200 μm, small pictures 100 μm).

**Figure 2 ijms-21-02151-f002:**
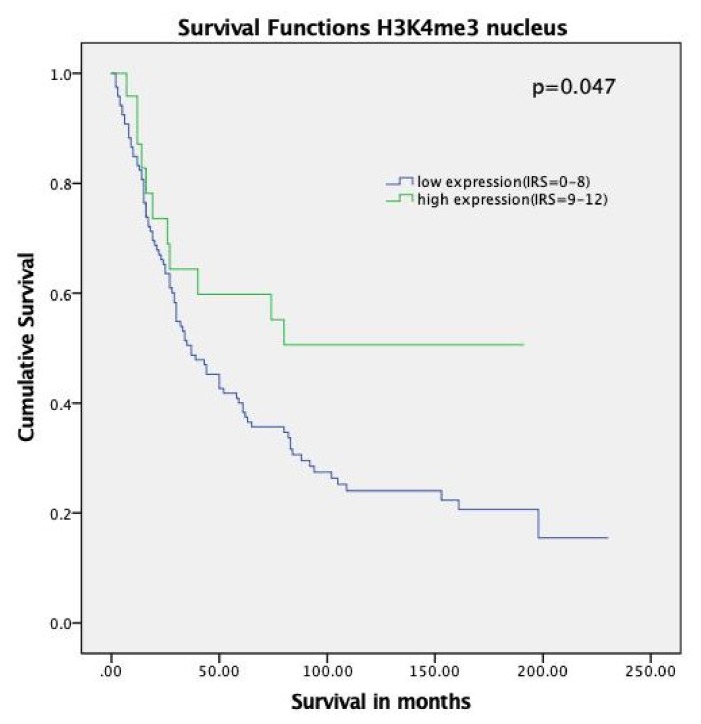
Kaplan–Meier analyses for overall survival: H3K4me3 (*p* = 0.047) with strong expression (Immunoreactive Score (IRS) = 9–12, green) compared to negative, weak, and moderate expression (IRS = 0–8, blue).

**Figure 3 ijms-21-02151-f003:**
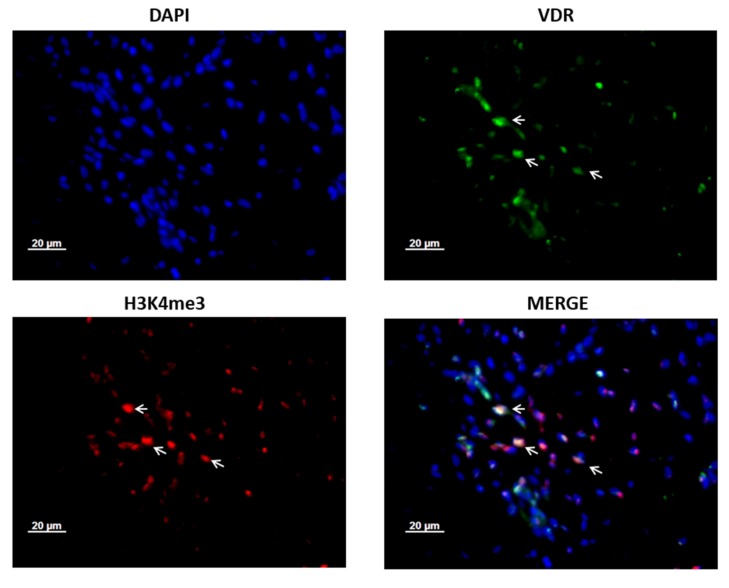
H3K4me3 is co-expressed with vitamin D receptor (VDR) in ovarian cancer patients’ tissue. Co-expression of VDR and H3K4me3 proteins is shown with →. Magnification × 40, scale bar = 20 μm. Tissues were co-stained with 4′,6-diamino-2-phenylindole (DAPI) (blue), H3K4me3 (red), and VDR (green).

**Figure 4 ijms-21-02151-f004:**
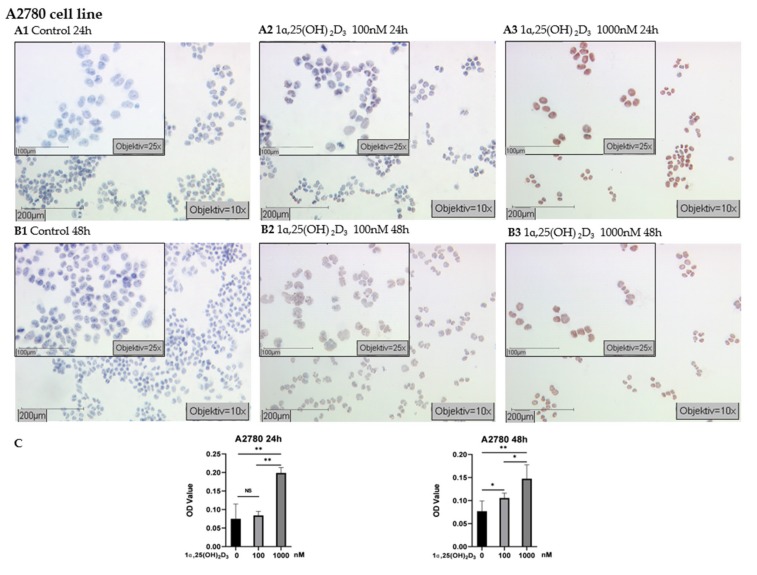
Detection of H3K4me3 with immunocytochemistry in A2780 cell line: (**A**) representative pictures of H3K4me3 immunocytochemistry staining of A2780 cells treated with 1α,25(OH)_2_D_3_ at different concentrations for 24 h (A1 control; A2 100 nM 1α,25(OH)_2_D_3_; A3 1000 nM 1α,25(OH)_2_D_3_); (**B**) representative pictures of H3K4me3 immunocytochemistry staining of A2780 cells treated with 1α,25(OH)_2_D_3_ at different concentrations for 48 h (B1 control; B2 100 nM 1α,25(OH)_2_D_3_; B3 1000 nM 1α,25(OH)_2_D_3_) (scale bars 200 μm, small pictures 100 μm); (**C**) ImageJ-based quantification of immunocytochemistry staining of H3K4me3 in A2780 cell line; NS, no statistical significance (*p* > 0.05); * with statistical significance (*p* < 0.05); ** with statistical significance (*p* < 0.01).

**Figure 5 ijms-21-02151-f005:**
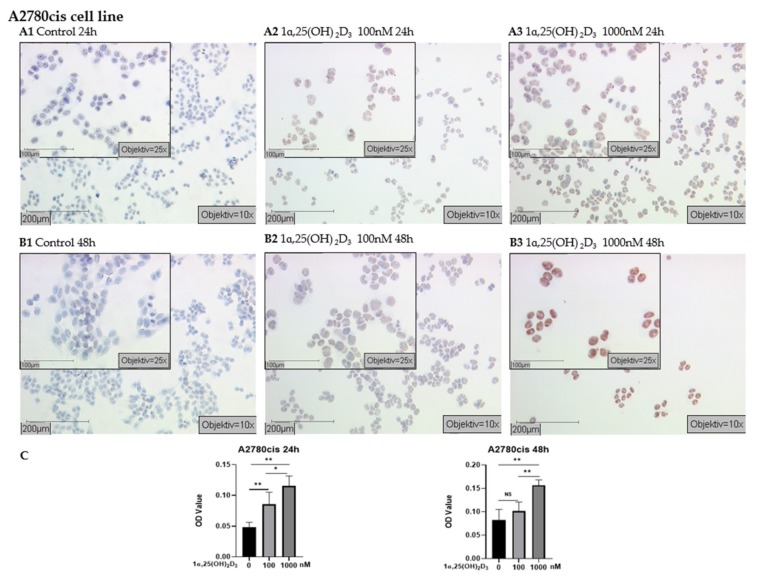
Detection of H3K4me3 with immunocytochemistry in A2780cis cell line: (**A**) representative pictures of H3K4me3 immunocytochemistry staining of A2780cis cells treated with 1α,25(OH)_2_D_3_ at different concentrations for 24 h (A1 control; A2 100 nM 1α,25(OH)_2_D_3_; A3 1000 nM 1α,25(OH)_2_D_3_); (**B**) representative pictures of H3K4me3 immunocytochemistry staining of A2780cis cells treated with 1α,25(OH)_2_D_3_ at different concentrations for 48 h (B1 control; B2 100 nM 1α,25(OH)_2_D_3_; B3 1000 nM 1α,25(OH)_2_D_3_) (scale bars 200 μm, small pictures 100 μm); (**C**) ImageJ-based quantification of immunocytochemistry staining of H3K4me3 in A2780cis cell line; NS, no statistical significance (*p* > 0.05); * with statistical significance (*p* < 0.05); ** with high statistical significance (*p* < 0.01).

**Figure 6 ijms-21-02151-f006:**
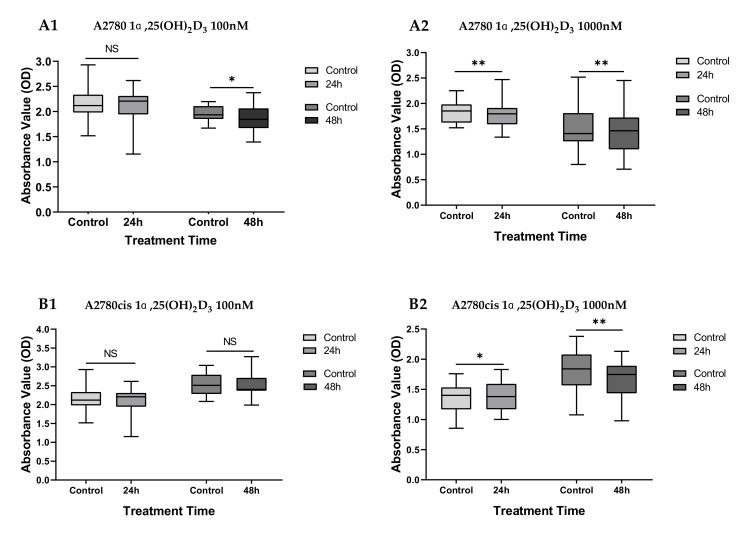
Effects of 1α,25(OH)_2_D_3_ treatment on cell proliferation in A2780 and A2780cis cell lines: Cell proliferation was measured by BrdU incorporation assay (optical density (OD) 450 nm). (**A1**) A2780 cells treated with 100n M 1α,25(OH)_2_D_3_ for 24 h and 48 h; (**A2**) A2780 cells treated with 1000 nM 1α,25(OH)_2_D_3_ for 24 h and 48 h; (**B1**) A2780cis cells treated with 100 nM 1α,25(OH)_2_D_3_ for 24 h and 48 h; (**B2**) A2780cis cells treated with 1000 nM 1α,25(OH)_2_D_3_ for 24 h and 48 h. NS, no statistical significance (*p* > 0.05); * with statistical significance (*p* < 0.05); ** with high statistical significance (*p* < 0.01), based on paired-samples T test.

**Table 1 ijms-21-02151-t001:** Expression profile of H3K4me3 staining regarding clinical and pathological characteristics.

Parameters	N		H3K4me3 Expression			*p*Value
		Negative	Weak	Moderate	High	
**Histology**						
serous	105	2	15	69	19	NS
clear cell	10	0	3	7	1	
endometrioid	20	1	3	7	2	
mucinous	12	2	3	6	2	
**Lymph node**						
pN0/X	97	5	14	63	15	NS
pN1	51	0	10	32	9	
**Overall Survival/months**						
<40.9 ≥40.9	79	2	16	51	10	NS
≥40.9	69	3	8	44	14	
**Grading**						
Low	33	3	6	24	5	NS
High	101	2	18	66	15	
**FIGO**						
I/II	41	3	6	24	8	NS
III/IV	108	2	18	53	16	
**Age/years**						
<64	70	3	13	42	12	NS
≥64	77	2	11	52	12	

NS = Not significant; FIGO = The International Federation of Gynecology and Obstetrics.

**Table 2 ijms-21-02151-t002:** Multivariate analysis.

Covariate	Coefficient (b_i_)	HR Exp(b_i_)	95% CI for Exp(B)		*p* Value
			Lower	Upper	
Histology (serous vs. others)	−0.096	0.91	0.70	1.18	0.458
Grade (low vs. high)	1.270	3.56	2.03	6.26	<0.001
FIGO (I, II vs. III, IV)	0.498	1.65	1.03	2.64	0.039
Patients’ age (<64 vs. ≥64 years)	−0.108	0.90	0.59	1.37	0.617
H3K4me3 (low vs. high)	−0.623	0.54	0.29	1.00	0.052

CI = confidence interval.

**Table 3 ijms-21-02151-t003:** IRS classification scoring systems.

Intensity of Staining	Percentage of Positive Cells	IRS (0–12)
0 = no color reaction	0 = no positive cells	0–1 = negative
1 = mild reaction	1 = <10% of positive cells	2–3 = weak
2 = moderate reaction	2 = 10–50% positive cells	4–8 = moderate
3 = intense reaction	3 = 51–80% positive cells	9–12 = strong positive
	4 = >80%positive cells	

IRS: Immunoreactive Score.

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
