# Peer review of "H3K4me3 Is a Potential Mediator for Antiproliferative Effects of Calcitriol (1α,25(OH)2D3) in Ovarian Cancer Biology"

_ijms, 2020, doi:10.3390/ijms21062151_

Round 1

Reviewer 1 Report

The Authors state that the studies were conducted according to the declaration of Helsinki which states that precaution must be taken to protect the privacy of research subjects and the confidentiality of their personal information. However the authors did not specifically state that sample were de-identified.

What controls were used for the immunohistochemical analysis? If no controls were used please explain the scientific rationale behind this.

Author Response

We truly appreciate the reviewers' comments and suggestions regarding our manuscript ijms-725339 “H3K4me3 is a potential mediator for antiproliferative effects of calcitriol (1ɑ, 25(OH)2D3) in ovarian cancer biology” and believe that it significantly improved following the H3K4me3 revision.

Please note that the page- and line-numbers below refer to the 'tracked changes' version of the revised manuscript.

Reviewer #1 (Comments to the Author (Required)):

The Authors state that the studies were conducted according to the declaration of Helsinki which states that precaution must be taken to protect the privacy of research subjects and the confidentiality of their personal information. However, the authors did not specifically state that sample were de-identified.

REPLY

We thank the reviewer for pointing out this lack of clarity in the manuscript added the about statement regarding de-identification of samples in the revised manuscript (p.12, ll.283-284.).

What controls were used for the immunohistochemical analysis? If no controls were used please explain the scientific rationale behind this.

REPLY

As positive and negative controls are crucial for interpretation of immunohistochemistry studies, these are implemented in our standard protocol and were also used in the present study. In the revised manuscript, supplementary figure 1 was added displaying staining for positive and negative controls from placenta (A) (p.4, ll109-111).

Reviewer 2 Report

The novelty is limited. Please note that the H3K4 is a well-known marker of ovarian cancer. (Oncogene. 2013.32:4586, J Cancer. 2019.10:4072.) Immunohistochemistry results (IHC) are not sufficient for final conclusion. H3K4 expression should be confirmed by western blot or qPCR. I recommend also to perform co-immunofluorescence detection of H3K4 and other ovarian cancer markers. Figures 1-4 have very low resolution in the manuscript.

Author Response

We truly appreciate the reviewers' comments and suggestions regarding our manuscript ijms-725339 “H3K4me3 is a potential mediator for antiproliferative effects of calcitriol (1ɑ, 25(OH)2D3) in ovarian cancer biology” and believe that it significantly improved following the H3K4me3 revision.

Please note that the page- and line-numbers below refer to the 'tracked changes' version of the revised manuscript.

Reviewer #2 (Comments to the Author (Required)):

The novelty is limited. Please note that the H3K4 is a well-known marker of ovarian cancer. (Oncogene. 2013.32:4586, J Cancer. 2019.10:4072.) Immunohistochemistry results (IHC) are not sufficient for final conclusion. H3K4 expression should be confirmed by western blot or qPCR.

REPLY

We appreciate the critical appraisal of our manuscript. In contrast to the mentioned references, our study first evaluated the global H3K4me3 expression and its relation to clinical outcome by immunohistochemistry (IHC) in ovarian cancer tissues.

As suggested, H3K4 as a protein has been described previously, but not its post-translational modification H3K4me3 in detail, which is new in the present manuscript. To best of our knowledge, our study is first to show the global expression of H3K4me3 and its prognostic role in ovarian cancer patients.

Both articles mentioned did not show the role and expression of global H3K4me3 in ovarian cancer patients directly. One is to examine bivalent chromatin domains containing H3K4me3 in only the high-grade serous ovarian cancer [1], and the other article is to study a member of H3K4me3 methyltransferase family in ovarian cancer. Neither of them did show the global expression of H3K4me3 protein and its relation to clinical outcomes. In this context, immunohistochemistry (IHC) is the best methods to determine the prognostic function of histone modification in cancer from our perspective.

Alterations in histone modification patterns have been shown to occur in cancer cells only at individual genes[2],they have not been associated with clinical outcome. Recently, histone modification patterns have been useful in distinguishing subtypes of cancer patients with distinct clinical outcomes, thereby expanding our prognostic capabilities[3].Heterogeneity in cellular (such as global or bulk) levels of histone modifications can be detected by IHC assay at the level of whole nuclei of cancer cells in tissue specimens[4]. In addition, methylation is controlled in a reversible fashion by methyltransferases and demethyltransferases. A kind of methyltransferase or demethyltransferases can have effect on several types of histone modifications [5].To indirectly determine expression and role of H3K4me3 in ovarian cancer via methyltransferases and demethyltransferases often obtains unclear results [6, 7].Therefore, most studies performed IHC to examine the global levels of H3K4me3 in cancer. Meanwhile, immunocytochemistry is also an option to evaluate histone H3 modification expression between normal and cancer cells [3, 8, 9].

In contrast, qPCR does not seem to be an adequate method to evaluate posttranslational modifications of a protein so that this approach would not add further information to our analysis.

To better illustrate the background of the described approach, we added the rationale for using IHC and ICC to detect histone modification and novelty of H3K4me3 in the introduction section of the revised manuscript (p.2, ll52-57 and p.2, ll64-67).

I recommend also to perform co-immunofluorescence detection of H3K4 and other ovarian cancer markers.

REPLY

We thank the reviewer for this valuable suggestion and agree that co-immunofluorescence might add important information to the manuscript. In this context, we decided to perform co-immunofluorescence of H3K4me3 and VDR in selected ovarian cancer specimens with known high expression of both factors as it has been shown that the level of histone modifications (including H3K4me3) significantly is modulated via enhancing genome-wide the rate of accessible chromatin and vitamin D receptor (VDR) binding by the 1,25(OH)2D3 stimulation[10].

This content was added in the Result section (p.7, ll127-135) and Method section (p.14, ll313-330) and discussion section (p.11, ll217-224).

Accordingly, numeration in the result (p.7, ll137) and (p.9, ll169) and method section (p.14, ll332, ll340; p.15, ll357, ll369) were adjusted.

Figures 1-4 have very low resolution in the manuscript.

REPLY

Updated figures with high resolution in the Result section were added to the revised manuscript.

References

[1] Chapman-Rothe N, Curry E, Zeller C, Liber D, Stronach E, Gabra H, et al. Chromatin H3K27me3/H3K4me3 histone marks define gene sets in high-grade serous ovarian cancer that distinguish malignant, tumour-sustaining and chemo-resistant ovarian tumour cells. Oncogene. 2013;32:4586-92.

[2] Jacobson S, Pillus L. Modifying chromatin and concepts of cancer. Curr Opin Genet Dev. 1999;9:175-84.

[3] Seligson DB, Horvath S, McBrian MA, Mah V, Yu H, Tze S, et al. Global levels of histone modifications predict prognosis in different cancers. Am J Pathol. 2009;174:1619-28.

[4] Seligson DB, Horvath S, Shi T, Yu H, Tze S, Grunstein M, et al. Global histone modification patterns predict risk of prostate cancer recurrence. Nature. 2005;435:1262-6.

[5] Pfau R, Tzatsos A, Kampranis SC, Serebrennikova OB, Bear SE, Tsichlis PN. Members of a family of JmjC domain-containing oncoproteins immortalize embryonic fibroblasts via a JmjC domain-dependent process. Proc Natl Acad Sci U S A. 2008;105:1907-12.

[6] Wang L, Mao Y, Du G, He C, Han S. Overexpression of JARID1B is associated with poor prognosis and chemotherapy resistance in epithelial ovarian cancer. Tumour Biol. 2015;36:2465-72.

[7] Jiang Y, Lyu T, Che X, Jia N, Li Q, Feng W. Overexpression of SMYD3 in Ovarian Cancer is Associated with Ovarian Cancer Proliferation and Apoptosis via Methylating H3K4 and H4K20. J Cancer. 2019;10:4072-84.

[8] Pedre X, Mastronardi F, Bruck W, Lopez-Rodas G, Kuhlmann T, Casaccia P. Changed histone acetylation patterns in normal-appearing white matter and early multiple sclerosis lesions. J Neurosci. 2011;31:3435-45.

[9] Crosio C, Heitz E, Allis CD, Borrelli E, Sassone-Corsi P. Chromatin remodeling and neuronal response: multiple signaling pathways induce specific histone H3 modifications and early gene expression in hippocampal neurons. J Cell Sci. 2003;116:4905-14.

[10] Nurminen V, Neme A, Seuter S, Carlberg C. The impact of the vitamin D-modulated epigenome on VDR target gene regulation. Biochim Biophys Acta Gene Regul Mech. 2018;1861:697-705.

Round 2

Reviewer 2 Report

The author answered all the questions without further comments.